# Real-World Insights into the Effectiveness and Tolerability of OnabotulinumtoxinA in Chronic Migraine: A Long-Term Evaluation of up to 11 Years

**DOI:** 10.3390/toxins17040208

**Published:** 2025-04-21

**Authors:** Antonio Santoro, Andrea Fontana, Massimiliano Copetti, Anna Maria Miscio, Giuseppe d’Orsi

**Affiliations:** 1Unit of Neurology, Fondazione IRCCS Casa Sollievo della Sofferenza, 71013 San Giovanni Rotondo, FG, Italy; anna.miscio@gmail.com (A.M.M.); g.dorsi@operapadrepio.it (G.d.); 2Unit of Biostatistics, Fondazione IRCCS Casa Sollievo della Sofferenza, 71013 San Giovanni Rotondo, FG, Italy; a.fontana@operapadrepio.it (A.F.); m.copetti@operapadrepio.it (M.C.)

**Keywords:** chronic migraine, onabotulinumtoxinA, real-world data, long-term outcomes, tailored therapy

## Abstract

Background: Chronic migraine (CM) is a debilitating neurological disorder that imposes substantial burdens on individuals and society, including diminished quality of life and increased healthcare utilization. While the efficacy of botulinum neurotoxin type A (BoNT-A) has been demonstrated in controlled trials, this longitudinal, real-world study offers unprecedented evidence of its long-term benefits, with patients followed for a median of 15 months (interquartile range: 6–36 months) and up to 11 years. Methods: This retrospective analysis included 579 patients diagnosed with CM who were newly treated with BoNT-A, according to the PREEMPT protocol, receiving injections every 12 weeks at doses of 155–195 units across 31–39 sites. Outcomes were assessed through changes in monthly headache days, frequency, symptomatic medication use, and migraine-related disability using Migraine Disability Assessment (MIDAS) scores up to 60 months from recruitment. Safety was evaluated by recording treatment-emergent adverse events (TEAEs), with a focus on long-term tolerability and subgroup variability. Results: Patients showed sustained improvements, with the mean number of monthly headache days decreasing from 22.7 to 5.5, and symptomatic medication use dropping from 33.4 to 3.7 mean doses at 60 months. Additionally, over 60% of patients improved from severe (MIDAS Grade IV) to minimal disability (MIDAS Grade I). Subgroup analysis revealed variability in response rates, emphasizing the need for personalized approaches. TEAEs were predominantly mild, with no new adverse events reported after 36 months, supporting the long-term safety of BoNT-A in real-world settings. Conclusions: This real-world study provides significant evidence for the long-term efficacy, safety, and tolerability of BoNT-A in the preventive treatment of CM. The findings highlight the importance of real-world data to account for patient variability and tailoring treatment strategies.

## 1. Introduction

Migraine, a debilitating and common neurological disorder characterized by recurrent episodes of headache, is increasingly recognized as a spectrum of diseases, with episodic and chronic forms representing different manifestations of the condition [1]. The mechanisms underlying the chronicity of headaches remain unclear. However, chronic migraine (CM), which affects approximately 2.5% of individuals worldwide, frequently evolves from episodic migraine [2]. Moreover, excessive medication use, defined as analgesic intake on 10–15 days per month, has been implicated in some cases [3]. CM is characterized by headaches occurring on at least 15 or more days/month for over 3 months, with at least 8 of these days meeting migraine-specific criteria, as defined by the International Classification of Headache Disorders, 3rd Edition (ICHD-3) [4]. Epidemiological studies estimate that the global prevalence of CM ranges between 0.9% and 5%, affecting millions of individuals and their families worldwide [5].

In Europe and America, CM is twice as prevalent in women compared to men, with rates ranging from 6% to 8% and from 15% to 18% (yearly prevalence), respectively [6]. Furthermore, the condition exhibits a bimodal age distribution, with prevalence peaking in the 18–29 and 40–49 age groups [6]. CM is the most debilitating migraine form; it profoundly impacts quality of life, daily activities and work productivity [7,8,9].

Therefore, CM is associated with higher direct medical costs, increased health care utilization and more comorbidity, including anxiety, depression, and medication overuse, contributing to disability even during headache-free periods [10,11,12]. Between therapeutic options now available, botulinum neurotoxin type A (BoNT-A) is a highly effective neuromodulator that inhibits muscle contraction by blocking the release of acetylcholine at the neuromuscular junction [13] and the release of calcitonin gene-related peptide (CGRP) [14]. This mechanism disrupts normal communication between nerves and muscles, effectively reducing muscle activity and modulating nerve signaling [15]. The Phase 3 research evaluating migraine prophylaxis therapy (PREEMPT 1 and PREEMPT 2) pivotal trials showed its efficacy and safety for CM prevention, demonstrating significant reductions in headache and migraine days [16,17]. BoNT-A has been shown to reduce monthly headache days by approximately 10 at 24 weeks, with nearly 50% of patients experiencing a ≥50% reduction in migraine days and improved quality of life [18]. Its favorable tolerability profile, characterized by low discontinuation rates and mild, transient adverse events, makes it a preferred long-term prophylactic option [19,20]. Clinical evidence further supports its effectiveness in reducing headache frequency, severity, and associated symptoms, while its ability to improve quality of life establishes it as a cornerstone therapy for managing and preventing CM [21,22].

Despite the results from clinical studies and real-life data, further evidence is needed to explore the long-term outcomes and detailed personalization of BoNT-A therapy [23,24]. This study aims to examine the long-term benefits and safety of BoNT-A in the management of CM, assess its impact in real-world clinical settings, and emphasize the importance of individualized treatment approaches tailored to the specific needs of each patient. The analysis is based on retrospectively collected data, up to 11 years, offering a comprehensive evaluation of its effectiveness and tolerability over an extended period.

## 2. Results

The study included 579 patients who initiated BoNT-A treatment for CM between March 2013 and May 2024. All patients underwent at least one post-treatment assessment at the Fondazione IRCCS Casa Sollievo della Sofferenza, San Giovanni Rotondo, Unit of Neurology. Out of the initially enrolled 618 patients, 39 were excluded due to treatment discontinuation after the first injection cycle, resulting in a final cohort of 579 participants. Figure 1 provides details on patient disposition, including the reasons for exclusions and discontinuations. Outcome measures were analyzed for up to 60 months following the last patient enrollment, as it was noted that no less than 10% of the recruited patients (66 patients, or 11.4%) continued receiving BoNT-A treatment up to the 60th month. By the end of the follow-up (60th month), 285 patients (49.2%) were still receiving treatment, while 294 (50.8%) had discontinued.

The study cohort had a mean age of 48.4 ± 13.8 (range: 14–88) years, with females accounting for 78% of the participants. Prior to enrollment, the average duration of chronic migraine history was 10.6 ± 9.7 (range: 1–60) years. Comorbidities included conditions such as anxiety and depression, which are frequently associated with CM. All patients were followed for a median of 15 months (interquartile range (IQR): 6–36 months). Baseline characteristics are detailed in Table 1.

### 2.1. Reduction in Headache Frequency

The analysis showed a significant and sustained reduction in headache frequency over time. At baseline, patients reported an average of 22.7 (95% Confidence Interval (CI): 22.0–23.4) headache days per month, which dropped significantly to 5.5 (95% CI: 4.5–6.4) days per month after 60 months of treatment (*p* for trend < 0.001). Similarly, the total monthly hours affected by headache decreased from 409.2 (95% CI: 393.2–425.7) to 24.5 (95% CI: 15.3–39.3) hours over the same period (*p* for trend < 0.001), reflecting a substantial reduction in overall headache burden. A progressive decrease in latency time following symptomatic drug administration was also observed, improving from 4.1 (95% CI: 3.9–4.2) hours at baseline to 1.6 (95% CI: 1.3–1.9) hours at 60 months (*p* for trend < 0.001), further emphasizing the therapeutic efficacy of the treatment. Moreover, a progressive decrease in visual analogue scale (VAS) was also observed, improving from 8.6 (95% CI: 8.4–8.8) at baseline to 5.6 (95% CI: 5.1–5.9) at 60 months.

A detailed analysis of temporal trends revealed that the most pronounced improvements occurred within the first 6–12 months of treatment, followed by a sustained stabilization of benefits up to the 60th month, without significant further reductions. This pattern is clearly depicted in Figure 2, which illustrates the progressive decline in headache frequency, symptomatic medication use, and latency time after symptomatic drug administration over the follow-up period. The persistence of these improvements over time suggests that BoNT-A therapy provides durable efficacy in the long-term management of chronic migraine.

Regarding patient responsiveness to BoNT-A, most participants demonstrated a significant reduction in headache burden. Specifically, 60.3% (n = 284) achieved a reduction of more than 75% in headache days during follow-up (high responders), 30.6% (n = 144) showed a reduction between 50% and 75% (responders) and 3.8% (n = 18) showed a reduction between 30% and 49.9% (partial responders). Only 5.3% (n = 25) were non-responders, highlighting the broad efficacy of BoNT-A in a heterogeneous patient population. These classifications highlight the variability of responses, though most patients experienced a clinically meaningful reduction in headache frequency and severity. Among the 66 patients who required continuous treatment up to the 60th month of follow-up, 40 (60.6%) and 61 (92.4%) of them were high responders in terms of headache days and hours, respectively. A similar trend was observed in the use of symptomatic medications, which decreased from an average of 33.4 (95% CI: 31.5–35.5) doses per month at baseline to 5.7 (95% CI: 3.7–8.3) doses per month at 60 months. This early and sustained reduction in symptomatic drug intake aligns with the decline in headache frequency and duration, reflecting a diminished overall need for acute symptom management. Patients exhibited variable responses to BoNT-A treatment, with outcomes classified based on the percentage reduction in headache days.

### 2.2. Long-Term Reduction in Migraine-Related Disability and Headache Burden

The analysis of migraine-related disability using the migraine disability assessment (MIDAS) scale showed that most patients (54.1%; n = 313) had severe impairment at baseline (Grade IV), while the remainder (45.8%; n = 265) exhibited moderate to severe limitations in their daily activities (Grades II–III) and only one patient (0.2%) had minimal disability (Grade I). Significant improvement was observed during the follow-up period. Of the 66 patients who were treated up to the 60th month, none remained in the severe disability category (Grade IV) and 62.1% (n = 41) regressed to minimal disability (Grade I). Additionally, 30.3% (n = 20) of these patients were classified as Grade II, while 7.6% (n = 5) remained in Grade III. These findings demonstrate the treatment’s effectiveness in significantly reducing migraine-related disability. The progressive shift in MIDAS grades over time is illustrated in Figure 3, showing the distribution of patients across all disability levels at each time point.

### 2.3. Adverse Events

The analysis of treatment-emergent adverse events (TEAEs) showed that a total of 32 events were observed in 579 patients, with an overall incidence rate of 3.0 events per 100 person-years (PY). Most of these events (87.5%, n = 28) were mild cases of neck pain, while 12.5% (n = 4) involved ptosis. The stratified analysis revealed that females experienced a higher incidence rate of TEAEs (3.6 events per 100 PY, n = 31/451) compared to males (0.5 events per 100 PY, n = 1/128) (Hazard Ratio (HR) = 0.13, 95% CI: 0.02–0.92 considering females as the reference group) (Figure 4). Notably, the last recorded event occurred at 36 months, and no further adverse events were reported beyond this time point. This finding suggests that the risk of adverse events decreases as treatment duration increases. Most adverse events occurred within the first year of treatment, followed by a sharp decline in incidence. The study results showed that the most reported adverse events, including neck discomfort and ptosis, were more common in female participants than in their male counterparts. No cases of transient neck weakness occurred.

A total of 294 patients (50.8%) discontinued BoNT-A treatment during the study. The most common reason for discontinuation was achieving a satisfactory clinical condition, either as defined by the clinician or as perceived by the patient (n = 174; 59.2% of discontinuers; median follow-up of 15 months). These patients were followed up after discontinuation and reported no significant worsening of their condition. Other reasons included pregnancy (n = 14; 4.8% of discontinuers; median follow-up of 13.5 months), initiation of an alternative preventive therapy such as CGRP receptor antagonists (n = 14; 4.8% of discontinuers; median follow-up of 39 months), and lack of efficacy (n = 7; 2.4% of discontinuers; median follow-up of 24 months). Additionally, 85 patients (28.9% of discontinuers; median follow-up of 3 months) were lost to follow-up. Among the patients who discontinued BoNT-A treatment, those who transitioned to alternative therapies were more likely to experience a TEAE than those who discontinued without initiating another treatment. This trend was most pronounced in patients who switched to CGRP. Patients in this group experienced a TEAE incidence rate of 8.4 events per 100 person-years (95% CI: 2.7–26.0), which was significantly higher than with other reasons for discontinuation. No adverse events were reported in patients who discontinued treatment due to pregnancy or lack of efficacy, further reinforcing the generally benign profile of the observed events (Table 2). Of the 66 patients who reached the 60th month, only 5 of them (7.6%) experienced a relapse.

## 3. Discussion

The findings of this study provide strong real-world evidence supporting the long-term effectiveness and safety of BoNT-A in the management of CM. A key finding is the rapid onset of therapeutic effects, with the most substantial improvements occurring within the first 6–12 months, followed by a sustained stabilization up to the 60th month. This trend was observed across all major clinical outcomes, including monthly headache days, symptomatic medication use, and latency time after symptomatic drug administration. Such an early and sustained response suggests that BoNT-A provides a fast and stable benefit, reducing the burden of CM without requiring frequent modifications in treatment strategy. Sample size calculation was not required in this study due to its exploratory nature. Indeed, the primary objective of this study was not to determine the minimum number of subjects required to detect a specific treatment effect within a given time frame, but rather to quantify trends in key measures of disease severity and treatment tolerability over the follow-up period. Our analysis demonstrated a significant and sustained reduction in headache frequency, from 22.7 days per month at baseline to 5.5 days after 60 months of treatment. These results align with data from the PREEMPT clinical program, which confirmed the efficacy of BoNT-A in reducing both headache and migraine days while improving overall quality of life [16,17]. Additionally, most patients treated with BoNT-A demonstrated a significant reduction in headache burden, with over 90% achieving a ≥50% reduction in headache days, while only 5.3% were classified as non-responders. These findings highlight a predominantly positive response in the study population, although the presence of a small subset of non-responders remains a critical aspect in the management of chronic migraine. Silberstein et al. (2024) reported that, even though some patients do not meet the conventional responder criterion (≥50% reduction in headache days), they may still experience benefits from BoNT-A treatment, such as reduced headache intensity or improved quality of life [25]. This discrepancy in response rates between studies could be attributed to methodological differences, variations in patient selection criteria, or differences in follow-up duration. However, both studies emphasize the importance of considering additional parameters beyond the reduction in headache days, such as improvements in disability and quality of life, to provide a more comprehensive assessment of treatment efficacy. BoNT-A also had a significant impact on migraine-related disability, as shown by the shift in MIDAS scores. At baseline, 54.1% of patients had severe impairment (Grade IV), while the remainder had moderate-to-severe limitations (Grades II–III). After 60 months of treatment, no patients remained in the severe disability category, with 62.1% transitioning to minimal disability (Grade I), 30.3% to Grade II, and 7.6% to Grade III. This improvement reflects not only a reduction in headache frequency but also an enhancement in daily functioning and work productivity, which is particularly relevant given the substantial impact of CM on social and professional life. Patients who adhered to regular treatment schedules experienced a progressive decline in symptomatic medication use, from 33.4 doses per month at baseline to 5.7 at 60 months. This reduction is crucial for medication overuse headache (MOH), a common challenge in CM management [26,27]. The long-term sustainability of these benefits further supports the role of continuous treatment with BoNT-A. These findings are consistent with previous long-term studies and real-world evidence, which suggest that sustained therapy improves clinical outcomes and provides durable relief in chronic migraine management [28,29]. Our findings confirm the favorable tolerability profile of BoNT-A, with a low incidence of TEAEs, predominantly mild and transient. The most reported adverse effects, such as neck pain and ptosis, occurred primarily during the initial months of treatment and decreased over time, corroborating data from the PREEMPT trials and other clinical studies [16,17,19]. Importantly, no new safety concerns emerged during the extended follow-up period, reinforcing the long-term safety profile of BoNT-A. The rapid onset of action observed in this study suggests that BoNT-A may provide early relief for CM patients, reducing symptom burden within the first year of treatment and maintaining these benefits over time. This underscores the importance of early patient engagement and adherence to treatment, as most of the therapeutic benefit is established within the initial treatment cycles. Approximately 50.8% of patients discontinued treatment, primarily due to perceived well-being (59.2%), while a smaller proportion transitioned to alternative therapies such as CGRP inhibitors. These findings emphasize the need for personalized treatment approaches, optimizing patient selection and adherence strategies to maximize long-term benefits.

However, several limitations should be considered when interpreting the results of this study. First, the retrospective nature of the study introduces potential selection bias and variability in the duration of follow-up. Secondly, the inclusion of patients with a history of previous preventive migraine treatment, such as antidepressants, calcium channel blockers, beta-blockers and antiepileptics, could represent a significant confounding factor. In addition, the lack of assessment of important confounding variables such as psychiatric conditions that may influence spontaneous remission of migraine is another notable limitation. Other limitations include the single-center design of this observational study, which may limit the generalizability of the results to other populations or clinical settings. The subjective definition of “well-being” as a reason for treatment discontinuation by both clinicians and patients also lacks standardized criteria. Finally, the exclusive use of the PREEMPT protocol for BoNT-A administration may limit the direct applicability of these results to other treatment regimens.

Despite the retrospective nature of this study, the large cohort and extended observation period provide valuable real-world insights.

As CM is the second leading cause of global disability, there is an urgent need for effective preventive therapies to mitigate its far-reaching clinical and societal impacts. Future prospective studies should focus on identifying predictors of treatment response and potential biomarkers that could guide personalized BoNT-A therapy for CM. These results solidify BoNT-A’s role as a cornerstone therapy for CM prevention, highlighting its protective actions against comorbidity risk factors, in line with findings from previous investigations and clinical guidelines [17,23,24,30,31].

## 4. Conclusions

This real-world study supports the long-term efficacy and safety of BoNT-A for CM prophylaxis up to 11 years. The treatment resulted in a rapid and substantial reduction in headache frequency and disability within the first 6–12 months, followed by a stable, long-lasting benefit up to 60 months. BoNT-A was well tolerated, with mostly mild and transient adverse events that declined over time. No new safety concerns emerged. These findings reinforce the role of BoNT-A as a cornerstone therapy for CM prevention, with a favorable safety profile and sustained clinical improvements over time.

## 5. Materials and Methods

This retrospective, observational study included patients with CM who received their first BoNT-A treatment between March 2013 and May 2024 and had at least one post-treatment assessment. The study was conducted at the Fondazione IRCCS Casa Sollievo della Sofferenza, San Giovanni Rotondo, in accordance with the Declaration of Helsinki and approved by the Ethics Committee of IRCCS Istituto Tumori Giovanni Paolo II of Bari at Fondazione IRCCS Casa Sollievo della Sofferenza of San Giovanni Rotondo (Protocol number: VI.0_07 APR 2015). Informed consent for the analysis and publication of data was obtained from all participants. Data integrity was ensured by cross-checking database entries against original records.

### 5.1. Study Subjects

Eligible participants were men and women aged ≥ 18 years with a diagnosis of CM, according to the ICHD-3 [4], and who completed at least one post-treatment assessment. Exclusion criteria included known allergies to botulinum toxin, a history of neurological disorders or deficit, uncontrolled systemic diseases, concomitant use of CGRP receptor antagonists or prior use of botulinum toxin for other indications. Women of childbearing potential were required to present a negative pregnancy test before each injection and to use reliable contraception.

To accurately reflect real-world practice, patients who were receiving concurrent preventive or symptomatic therapies were not excluded from the study.

### 5.2. Treatment Protocol

BoNT-A (BOTOX^®^, Allergan plc, Dublin, Ireland) was administered every 12 weeks following the PREEMPT protocol, with doses ranging from 155 to 195 units injected across 31–39 sites. For patients with MOH, a six-day detoxification regimen with intramuscular betamethasone preceded BoNT-A initiation. Post-detoxification, patients were advised to limit non-steroidal anti-inflammatory drugs use to a maximum of two doses per week. Patients who were lost to follow-up were analyzed up to their last recorded visit whereas those who discontinued the BoNT-A treatment were analyzed up to the date of discontinuation. Reasons for discontinuation included clinician-defined well-being, lack of efficacy, personal choice, pregnancy, or switching to other therapies (e.g., CGRP inhibitors) due to lack of efficacy of botulinum toxin or by personal choice.

### 5.3. Outcome Measurement

Outcome measures were analyzed bi-annually, focusing on changes in headache frequency (both days and hours), acute medication use, latency time, and MIDAS grade distribution from baseline (T0) to each subsequent time point. Efficacy was assessed at baseline (T0) and at six-month intervals. Outcomes were assessed within a timeframe that ensured that at least 10% of the original study cohort remained. This arbitrary choice was made to ensure the stability of the estimates. Primary outcomes included: headache frequency: monthly headache days and hours; acute medication use: number of symptomatic drug doses and latency time post-administration; disability: MIDAS scores and their categorical grade distribution. Safety was evaluated by recording TEAEs, categorized by type and severity, with incidence rates calculated per 100 person-years. Responsiveness to BoNT-A treatment was calculated by measuring the percentage reduction in headache days and hours, with patients categorized into four groups based on response: non-responders (<30% reduction), partial responders (≥30% and ≤49.9%), responders (≥50% and ≤75%), and high responders (>75%).

### 5.4. Statistical Analysis

Continuous variables are reported as means ± SD, medians with IQR, and range, while categorical variables are reported as frequencies and percentages. For categorical comparisons, the Chi-square or Fisher’s exact test was used. Changes over time were analyzed using hierarchical generalized linear models (HGLMs), with follow-up time as a categorical covariate. Poisson and binomial distributions were assumed for count and binary outcomes, respectively. Count outcomes included the number of headache days/hours per month, VAS score, symptomatic drug use, and drug latency hours, while binary outcomes included MIDAS scale categories and treatment responsiveness groups (both treated as dummy variables). Repeated measures were accounted for using a first-order autoregressive covariance structure. HGLMs provided estimates of means (or percentages for binary data), along with their 95% CIs. The presence of a linear trend in the measures over time was assessed by including follow-up time as a continuous covariate. TEAEs rates were expressed as the number of events per 100 person-years, with confidence intervals for these incidence rates computed using Poisson regression. This model was also used to compare TEAE rates between patients who continued treatment and those who discontinued, and to calculate HR, using ongoing patients as the reference group. Additionally, Kaplan–Meier curves illustrating the survival probability of not experiencing any adverse events during follow-up were presented for the total cohort and by gender. A *p*-value of <0.05 indicates statistical significance. All statistical analyses were performed using SAS software (release 9.4, SAS Institute, Cary, NC, USA), and plots were generated using R software (version 4.3).

## Figures and Tables

**Figure 1 toxins-17-00208-f001:**
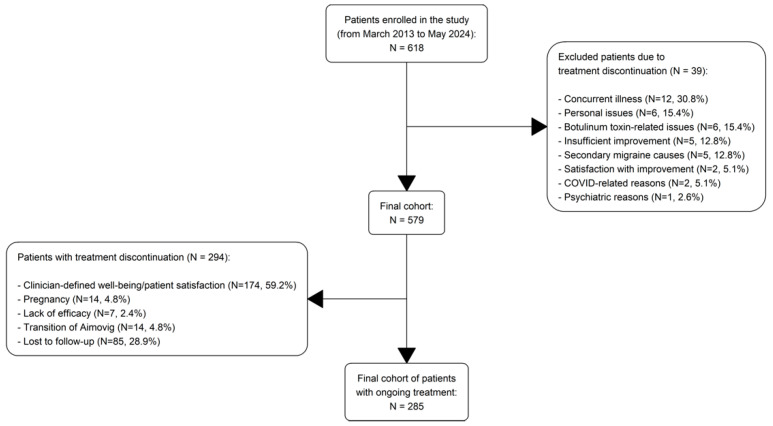
Patient disposition flow diagram.

**Figure 2 toxins-17-00208-f002:**
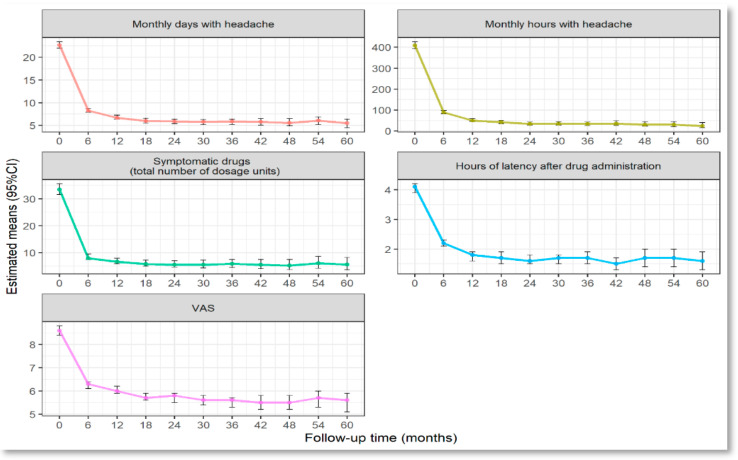
Plots of estimated mean values for monthly headache days, hours, number of symptomatic drugs received, latency time after symptomatic drug administration, VAS means at baseline and after 60 months, assessed every 6 months using hierarchical generalized linear models. Error bars represent 95% confidence intervals (CI) around the estimated means. At each time point (T), the numbers of patients were as follows: T0 = 579, T6 = 471, T12 = 377, T18 = 281, T24 = 231, T30 = 191, T36 = 153, T42 = 118, T48 = 99, T54 = 82 and T60 = 66.

**Figure 3 toxins-17-00208-f003:**
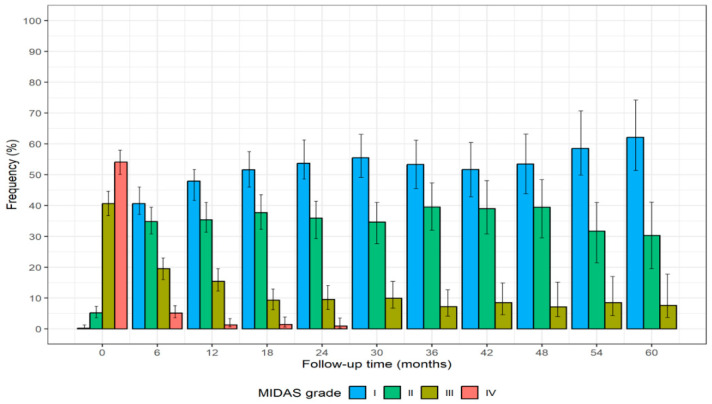
Bar plots for MIDAS grade at baseline and after 60 months, evaluated at each 6 months and estimated using hierarchical generalized linear models. Error bars represent 95% confidence interval (CI) around the estimated proportions. At each time point (T), the numbers of patients were as follows: T0 = 579, T6 = 471, T12 = 377, T18 = 281, T24 = 231, T30 = 191, T36 = 153, T42 = 118, T48 = 99, T54 = 82 and T60 = 66.

**Figure 4 toxins-17-00208-f004:**
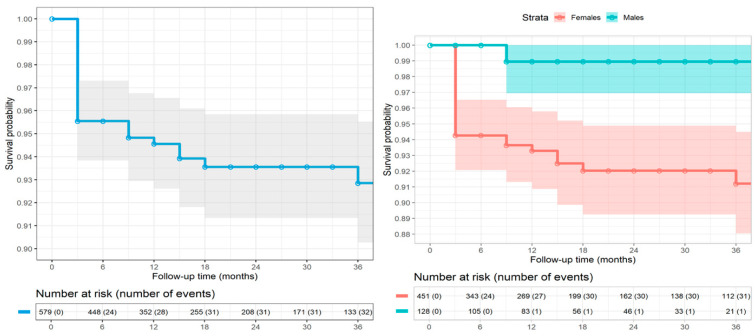
Kaplan–Meier curves of the rate of treatment-emergent adverse events (i.e., 28 patients with neck pain and 4 with ptosis) in the overall cohort (**left**) and by sex (**right**). The table below each graph shows the number of subjects at risk and the cumulative number of adverse events over time. As the last event occurred at 36 months, the timeline (x-axis) ends at this point.

**Table 1 toxins-17-00208-t001:** Clinical characteristics at baseline, in the overall cohort and by sex.

		Overall (N = 579)	Males (N = 128)	Females (N = 451)	*p*-Value
Age (years)	Mean ± SD	48.4 ± 13.8	48.1 ± 13.6	48.5 ± 13.8	0.814 °
	Median (IQR)	49 (39–57)	49 (40–56.5)	48 (39–58)	
	Range	14–88	14–83	16–88	
Age < 50 years	n (%)	313 (54.0)	69 (53.9)	244 (54.1)	0.749 ^#^
Age 50–65 years	n (%)	202 (34.9)	47 (36.7)	155 (34.4)	
Age > 65 years	n (%)	64 (11.1)	12 (9.4)	52 (11.5)	
Years of chronic headache	Mean ± SD	10.6 ± 9.7	10.4 ± 9.4	10.6 ± 9.7	0.992 *
	Median (IQR)	8 (3–15)	7 (4–15)	9 (3–15)	
	Range	1–60	1–45	1–60	
Presence of photophobia	n (%)	445 (76.9)	85 (66.4)	360 (79.8)	0.002 ^#^
Presence of phonophobia	n (%)	443 (76.5)	81 (63.3)	362 (80.3)	<0.001 ^#^
Worsens with movement	n (%)	493 (85.1)	100 (78.1)	393 (87.1)	0.011 ^#^
Presence of allodynia	n (%)	173 (29.9)	27 (21.1)	146 (32.4)	0.014 ^#^
Intramuscular therapy use	n (%)	120 (20.7)	19 (14.8)	101 (22.4)	0.063 ^#^
Intravenous therapy use	n (%)	43 (7.4)	8 (6.3)	35 (7.8)	0.565 ^#^
Prophylactic therapy use	n (%)	197 (34.0)	35 (27.3)	162 (35.9)	0.071 ^#^
Use of NSAIDs	n (%)	411 (71.0)	94 (73.4)	317 (70.3)	0.473 ^§^
Use of Triptans	n (%)	235 (40.6)	50 (39.1)	185 (41.0)	0.691 ^#^
Use of other drugs	n (%)	126 (21.8)	20 (15.6)	106 (23.5)	0.068 ^#^
Follow-up (months)	Mean ± SD	24.8 ± 24.3	20.8 ± 19.7	25.9 ± 25.4	0.214 *
	Median (IQR)	15 (6–36)	15 (7.5–30)	15 (6–39)	
	Range	0–123	0–108	0–123	

SD: standard deviation; IQR: interquartile range (i.e., first–third quartiles); NSAIDs: Non-steroidal anti-inflammatory drugs; ^#^
*p*-value from Chi-Square test; ^§^
*p*-value from Fisher exact test; ° *p*-value from two-sample *t*-test; * *p*-value from Mann–Whitney U test.

**Table 2 toxins-17-00208-t002:** Rate of treatment-emergent adverse events in the total cohort, in continuing patients and in discontinuing patients, subdivided by reason for discontinuation. PY: person-years; IR: incidence rate, defined as number events per 100 PY; CI: confidence interval; HR: hazard ratio; NA: not available; NE: not estimable as no events were recorded; * the last adverse event occurred at 36 months. Of the 32 events recorded, 28 (87.5%) were related to neck pain, while the remaining 4 events (12.5%) were cases of ptosis.

	N. Subjects	N. Events *	PY	IR (95% CI)	HR (95% CI)	*p*-Value
All patients (total cohort)	579	32	1081.3	3.0 (2.1–4.2)	NA	NA
Ongoing patients	285	17	675.3	2.5 (1.6–4.1)	Ref. group	NA
Discontinuers due to wellbeing	174	11	277.3	4.0 (2.2–7.2)	1.58 (0.74–3.36)	0.240
Discontinuers due to pregnancy	14	0	26.5	0.0 (NE)	NE	NA
Discontinuers due to lack of efficacy	7	0	16.5	0.0 (NE)	NE	NA
Discontinuers due to starting Aimovig	14	3	35.8	8.4 (2.7–26.0)	3.33 (0.98–11.38)	0.054
Lost to follow-up	85	1	50.0	2.0 (0.3–14.2)	0.79 (0.11–5.97)	0.822

## Data Availability

The data presented in this study are available on request from the corresponding author as they contain information that could compromise the privacy of research participants.

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
