# Peer review of "Real-World Insights into the Effectiveness and Tolerability of OnabotulinumtoxinA in Chronic Migraine: A Long-Term Evaluation of up to 11 Years"

_toxins, 2025, doi:10.3390/toxins17040208_

Round 1
Reviewer 1 Report
Comments and Suggestions for Authors
This is a useful report of the effects of BNTxA on chronic migraine headache. The number of patients covered is impressive and the details of followup are impressive. The outcome is persuasive of a significantly beneficial effect.
I have a few comments/suggestions to make. First of all, in this day and age of widespread use of CGRP inhibitors in the treatment of migraine, the authors should be very clear in the methods section if the subjects used any CGRP inhibitors along with botulinum toxin. Secondly, a rather large number of subjects stopped the treatment apparently because of the absence of headache. No further comment is made about these subjects. Do the authors have any followup on these patients after they stopped treatment? Did they continue to be headache free or to have minimal trouble with migraine? Or did they develop chronic migraine again and use a different preventative treatment such as CGRP inhibitors. This would be or interest.
Do the authors know why some subjects moved on to CGRP inhibitors? Was this because botulinum toxin was less effective, or inconvenient? Was there an overlap of using both treatments before making the switch?
The authors state that the PREEMT protocol was used, presumable of fixed injection sites rather that 'follow the pain". Do the authors have any comments for either the introduction or the discussion about the effectiveness of the two different injection techniques?
Regarding adverse side effects, the authors report that the most common side effect was neck pain. Were there any cases of transient neck weakness?
Some specific comments related to the manuscript: line 14: headache days frequency is an awkward term. Headache days is clear and may be different from headache frequency as one headache could encompass several or more days. Perhaps the authors meant "headache days, frequency..." and simply left out the comma?
In the introduction, line 37 should have a period instead of a semi-colon.
Line 46: women are three times more likely than men to have chronic migraine, but you quoted percentages shows only twice as likely.
Line 55-57: the authors state that acetylcholine release is blocked by botulinum toxin, suggesting that this is the only neurotransmitter that is affected. However, CGRP release is also blocked and that is relevant for migraine. [See Toxins 2024;16:309; Toxins 2024;16:431; Pereira et al. J Neural Transm 2025; Mar 4.doi:10.1007/s00702-024-w]
Line 57: sentence is awkward and not clear
Figure 3 is confusing as to what you are trying to say. In fact, the text is very clear and the figure offers no greater understanding.
Likewise, figure 6 does not add any greater understanding to the text and could be eliminated.
Author Response
Comments 1: This is a useful report of the effects of BNTxA on chronic migraine headache. The number of patients covered is impressive and the details of followup are impressive. The outcome is persuasive of a significantly beneficial effect.
Response 1: We thank the Reviewer for the overall positive feedback.
I have a few comments/suggestions to make.
Comments 2: First of all, in this day and age of widespread use of CGRP inhibitors in the treatment of migraine, the authors should be very clear in the methods section if the subjects used any CGRP inhibitors along with botulinum toxin.
Response 2: None of the 579 patients initiated concomitant CGRP inhibitors along with botulinum toxin. This clarification has now been added to the Study Subjects methods section. Only 14 patients (2.4%) discontinued botulinum toxin treatment during follow-up and began using a CGRP inhibitor as an alternative new treatment. The follow-up period for these patients concluded at the time of discontinuation.
Comments 3: Secondly, a rather large number of subjects stopped the treatment apparently because of the absence of headache. No further comment is made about these subjects. Do the authors have any followup on these patients after they stopped treatment? Did they continue to be headache free or to have minimal trouble with migraine? Or did they develop chronic migraine again and use a different preventative treatment such as CGRP inhibitors. This would be or interest.
Response 3: The majority of patients who stopped treatment did so because they felt better. These patients were followed up after discontinuation and reported no significant worsening of their condition. This patient group will be the subject of a new study (we will examine the long-term outcomes of these patients in further research).
Comments 4: Do the authors know why some subjects moved on to CGRP inhibitors? Was this because botulinum toxin was less effective, or inconvenient? Was there an overlap of using both treatments before making the switch?
Response 4: The 14 patients who switched to CGRP inhibitors did so due to either the ineffectiveness of botulinum toxin or by personal choice. There was no overlap of treatments; they did not use both simultaneously. This clarification has now also been added to the methods section.
Comments 5: The authors state that the PREEMT protocol was used, presumable of fixed injection sites rather that 'follow the pain". Do the authors have any comments for either the introduction or the discussion about the effectiveness of the two different injection techniques?
Response 5: We have no comment on this as each patient received a fixed dose (155 to 195 units) at 31 to 39 injection sites.
Comments 6: Regarding adverse side effects, the authors report that the most common side effect was neck pain. Were there any cases of transient neck weakness?
Response 6: No cases of transient neck weakness occurred.
Some specific comments related to the manuscript:
Comments 7: line 14: headache days frequency is an awkward term. Headache days is clear and may be different from headache frequency as one headache could encompass several or more days. Perhaps the authors meant "headache days, frequency..." and simply left out the comma? In the introduction, line 37 should have a period instead of a semi-colon.
Response 7: We apologize for the oversight. The term "headache days frequency" was a typo error, and the text has now been corrected accordingly. Additionally, we have corrected the semicolon in Line 37 to a period.
Comments 8: Line 46: women are three times more likely than men to have chronic migraine, but you quoted percentages shows only twice as likely.
Response 8: The Reviewer is right; this was another typo error. In fact, studies in Europe and America report that women are twice as likely as men to have chronic migraine. We have now corrected the typo and updated the percentages to match those reported by Katsarava et al. (see reference n.6).
Comments 9: Line 55-57: the authors state that acetylcholine release is blocked by botulinum toxin, suggesting that this is the only neurotransmitter that is affected. However, CGRP release is also blocked and that is relevant for migraine. [See Toxins 2024;16:309; Toxins 2024;16:431; Pereira et al. J Neural Transm 2025; Mar 4.doi:10.1007/s00702-024-w]. Line 57: sentence is awkward and not clear.
Response 9: We appreciate the Reviewer’s comment. We have clarified that botulinum toxin (BoNT-A) is a highly effective neuromodulator that inhibits muscle contraction by blocking not only the release of acetylcholine but also the release of CGRP and other neurotransmitters (substance P, glutamate, etc.). The awkward sentence in Line 57 has now been removed.
Comments 10: Figure 3 is confusing as to what you are trying to say. In fact, the text is very clear and the figure offers no greater understanding. Likewise, figure 6 does not add any greater understanding to the text and could be eliminated.
Response 10: We agree that Figures 3 and 6 do not add significant value and have removed them from the manuscript.
Reviewer 2 Report
Comments and Suggestions for Authors
important followup paper
Author Response
Comments 1: Important followup paper
Response 1: We thank the Reviewer for the overall positive feedback.
Reviewer 3 Report
Comments and Suggestions for Authors
The study "Real-World Insights into the Long-Term Effectiveness and Tolerability of OnabotulinumtoxinA in Chronic Migraine: An 11-Year Evaluation" is an interesting study reflecting a current and relevant topic. However, I have some serious concerns about the study. The title of the study immediately presents misleading information. I expected an 11-year study, but then I realized the follow-up period is 5 years (60 months). In the end, only 66 patients reached the 60-month mark, while after one year, 377 of the initial 579 patients remained. As a result, two points immediately come to mind:
- The methodology states that there was a baseline (T0) and measurements every 6 months, but only in the results section does it mention that the follow-up period ends after 60 months. This information is confusing. The study is not clear about the timeframes or its termination.
- There was no analysis of the sample size power for the results. Can I trust the final 60-month timeframe with only 66 patients? Did these 66 patients stay in the study because they were relapse cases?
- In my opinion, it could be considered a study with 66 patients over 60 months. Wouldn’t it have been more appropriate to consider a study endpoint, for example, at 24 months with more patients to give more confidence to the results? Additionally, the 11 years should be removed from the title, and mentioning "11 years of follow-up" in the abstract is misleading.
- The inclusion and exclusion criteria are confusing. Could the patients have undergone other treatments that contributed to reducing pain and migraines? How can we be sure it was the botulinum toxin? What are the confounding variables? What could have contributed to the bias in the study?
- Most patients dropped out of the study because they felt better. This leads to the question: Shouldn’t the study have focused on understanding how many patients, over 11 years, needed continuous treatment every 12 weeks? What was the need for the frequency of treatment? I think this question would have been relevant. And if they repeated treatments, was it because they had recurrences? What was the timeframe of the 174 patients who left the study because they were doing well?
- In the discussion, I think the authors could have cited others on the topic. Considering the limitations already mentioned in the study design, it is crucial to acknowledge them.
I don’t think this article is ready for publication for these reasons. The article’s narrative should be reconsidered. I believe it would be more accurate to say, "Over 11 years, 579 patients underwent treatment, and after a certain period, patients left the study because they were doing better, etc." Each year, an evaluation should be conducted to understand why patients continued in the study, allowing an assessment of the need to continue treatment.
Comments on the Quality of English LanguageThe English is understandable, with only a few minor inaccuracies, such as the acronym MOH not being in full the first time it appears. I think a little revision would improve this.
Author Response
The study "Real-World Insights into the Long-Term Effectiveness and Tolerability of OnabotulinumtoxinA in Chronic Migraine: An 11-Year Evaluation" is an interesting study reflecting a current and relevant topic. However, I have some serious concerns about the study. The title of the study immediately presents misleading information. I expected an 11-year study, but then I realized the follow-up period is 5 years (60 months). In the end, only 66 patients reached the 60-month mark, while after one year, 377 of the initial 579 patients remained. As a result, two points immediately come to mind:
Comments 1: The methodology states that there was a baseline (T0) and measurements every 6 months, but only in the results section does it mention that the follow-up period ends after 60 months. This information is confusing. The study is not clear about the timeframes or its termination.
Response 1: The Reviewer is right. We did not make it clear that the outcomes were assessed within a timeframe that ensured that at least 10% of the original study cohort remained. This arbitrary choice was made to ensure the stability of the estimates. Specifically, we found that 60 months was the ideal timeframe as only 66 patients out of 579 (11.4%) remained in the study.
Comments 2: There was no analysis of the sample size power for the results. Can I trust the final 60-month timeframe with only 66 patients? Did these 66 patients stay in the study because they were relapse cases?
Response 2: Sample size calculation was not required in this study due to its exploratory nature. The primary objective was not to determine the minimum number of subjects required to detect a specific treatment effect within a given time frame, but rather to quantify trends in key measures of disease severity and treatment tolerability over the follow-up period. This clarification has now also been added to the discussion section. The 66 patients who remained after 60 months were those for whom botulinum toxin did not have an immediate effect and required continued treatment. Only 5 of these 66 patients experienced a relapse.
Comments 3: In my opinion, it could be considered a study with 66 patients over 60 months. Wouldn’t it have been more appropriate to consider a study endpoint, for example, at 24 months with more patients to give more confidence to the results? Additionally, the 11 years should be removed from the title, and mentioning "11 years of follow-up" in the abstract is misleading.
Response 3: We respectfully disagree with the Reviewer’s suggestion to focus only on patients who were followed for 24 or 60 months, as this would result in the loss of valuable information about those who completed the study much earlier. However, we fully agree that the title is misleading, and we have revised it accordingly: "Real-World Insights into the Effectiveness and Tolerability of OnabotulinumtoxinA in Chronic Migraine: A Long-Term Evaluation of Up to 11 Years." We have also made the appropriate changes in the abstract to reflect this revised title.
Comments 4: The inclusion and exclusion criteria are confusing. Could the patients have undergone other treatments that contributed to reducing pain and migraines? How can we be sure it was the botulinum toxin? What are the confounding variables? What could have contributed to the bias in the study?
Response 4: All recruited patients were using BoNT-A (BOTOX®; Allergan plc, Ireland) continuously as their only treatment to reduce pain and migraine frequency. Some patients had used other treatments prior to enrolment, such as antidepressants, calcium channel blockers, beta-blockers and antiepileptic drugs. The most important confounding variables could be psychiatric conditions that could affect spontaneous recovery from migraine, but unfortunately these factors were not assessed in this study.
Comments 5: Most patients dropped out of the study because they felt better. This leads to the question: Shouldn’t the study have focused on understanding how many patients, over 11 years, needed continuous treatment every 12 weeks? What was the need for the frequency of treatment? I think this question would have been relevant. And if they repeated treatments, was it because they had recurrences?
Response 5: The study was not focused on determining how many patients needed continuous treatment. The protocol we followed, PREEMT, requires that patients receive treatment every 12 weeks. They were treated until they had fewer than 4 headache days per month for at least one year. It is also possible that a patient, despite having fewer than 4 headache days per month, chooses to continue treatment voluntarily.
Comments 6: What was the timeframe of the 174 patients who left the study because they were doing well?
Response 6: The median timeframe for these 174 patients who left the study because they were doing well was 16 months (as shown in the boxplot in the former Figure 6).
Comments 7: In the discussion, I think the authors could have cited others on the topic. Considering the limitations already mentioned in the study design, it is crucial to acknowledge them.
Response 7: We believe that the discussion is already quite detailed, and we have made sure to acknowledge all the study's limitations.
Comments 8: I don’t think this article is ready for publication for these reasons. The article’s narrative should be reconsidered. I believe it would be more accurate to say, "Over 11 years, 579 patients underwent treatment, and after a certain period, patients left the study because they were doing better, etc." Each year, an evaluation should be conducted to understand why patients continued in the study, allowing an assessment of the need to continue treatment.
Response 8: We appreciate the Reviewer’s constructive feedback. We have revised the narrative to more clearly reflect the progression of patients over the course of the study, noting that 579 patients were initially treated and that over time many patients left the study due to improvements in their condition. We will also highlight the importance of evaluating why patients stayed in the study to better understand the need for continued treatment.
Comments 9: The English is understandable, with only a few minor inaccuracies, such as the acronym MOH not being in full the first time it appears. I think a little revision would improve this.
Response 9: We apologize for the oversight. The acronym MOH has now been specified and the English has been revised.
Round 2
Reviewer 3 Report
Comments and Suggestions for Authors
The authors gave satisfactory answers to all the questions raised.